# Risk Analysis of Heavy Metals Migration from Sewage Sludge of Wastewater Treatment Plants

**DOI:** 10.3390/ijerph191811829

**Published:** 2022-09-19

**Authors:** Robert Kowalik, Jarosław Gawdzik, Paulina Bąk-Patyna, Piotr Ramiączek, Nebojša Jurišević

**Affiliations:** 1Faculty of Environmental, Geomatic and Energy Engineering, Kielce University of Technology, 25-314 Kielce, Poland; 2Faculty of Civil Engineering and Architecture, Kielce University of Technology, 25-314 Kielce, Poland; 3Department for Energy and Process Engineering, Faculty of Engineering, University of Kragujevac, 34000 Kragujevac, Serbia

**Keywords:** heavy metals, sewage sludge, mobility, environmental pollution

## Abstract

More and more attention in sewage sludge management is being devoted to its environmental utilization. This approach is justified both from economic and environmental points of view. However, as with any method, there are certain possibilities and limitations. The goal of the natural utilization of sewage sludge is to recover the valuable agronomic properties and fertilizing potential of the sludge. The main aspect limiting the possibility of using sludge as a fertilizer is the heavy metal content. In this paper, an analysis of the risk of environmental contamination in the case of application of sewage sludge with different forms of sludge treatment was carried out. Risk indices such as Igeo and PERI, based on the comparison of total metal content in sludge and soil, as well as RAC and ERD indices, which take into account the mobility of metals in soil, were calculated. It was shown that high levels of potential risk and geoaccumulation indicators do not necessarily disqualify the use of sewage sludge, the key aspect is the form of mobility in which the heavy metals are found in the sludge, and this should be the only aspect taken into account for the possibility of their environmental use.

## 1. Introduction

The progress of civilization in the last century has contributed significantly to the improvement of human life quality. As a result of these changes, the water demand and pollutant load of discharged wastewaters have increased [1,2,3]. It is, therefore, necessary to continuously expand and modernize the infrastructure for wastewater collection and treatment. However, this contributes to an increase in the amount of sewage sludge generated as a byproduct of sewage treatment plant processes. So far, no waste-free method or effective solution has been developed to completely eliminate sewage sludge from the environment [4,5]. However, there are many methods to utilize the sludge generated this way. Agricultural use of sludge fertilizers is particularly beneficial due to its high soil-forming and fertilizing properties [6,7,8]. To this end, the method of sewage sludge management is primarily determined by the amount and properties of the sludge [9,10]. Sludge with high reclamation fertilizer values can be used as an organic fertilizer as long as the micropollutant content does not have a negative impact on the soil environment [11,12,13]. The permissible levels of heavy metals in the application of sewage sludge in Poland and the world are listed in Table 1.

The chemical forms of metals present in sewage sludge can be identified by sequential extraction or speciation based on the fractionation of compounds. The use of this analytical procedure ensures the separation of the test material into fractions characterized by different degrees of mobility [19].

Soluble metals, which are highly mobile and readily available, pose the greatest threat to soil inhabitants as micronutrients that enter ground and surface waters move up the trophic chain. Heavy metals in the soil are not immediately absorbed by plants; however, they can slowly form hazardous solutions over time [20]. Some essential elements, such as Fe, Co, Cu, Cr, Mo, Mn, Se, Ni, and Zn, are required for organisms in trace amounts; however, they become toxic at higher levels. Nonessential elements such as Sb, Pb, Hg, Ag, and As are toxic and not needed by living organisms [21]. However, most wastewaters and wastes contain heavy metals in amounts sufficient to cause toxicity to crop plants [22].

Sewage sludge, a byproduct of wastewater treatment, can be managed in several ways. However, the most favorable variant from an ecological standpoint and in terms of a circular economy is its use for agricultural purposes.

Municipal sewage sludge can be used as a substrate for the production of organic fertilizers or plant growth aids, but the most important criterion it must meet is the total content of heavy metals. In that regard, the goal of this study is to confirm that a high concentration of heavy metals in the sludge does not always rule out the possibility of sludge agricultural use. The key, therefore, is the content of metals in fractions that tend to migrate deep into the environment and, thus, can easily enter the food chain.

This study investigates the sewage sludge content of four wastewater treatment plants in Poland using different wastewater treatment technologies. The investigation considers heavy metal concentrations, mobility, and the risk of contamination of the environment. Based on the results, the geoaccumulation index (Igeo), potential environmental risk index (PERI), risk assessment code (RAC), and environmental risk determinant (ERD) were calculated. All indicators were then compared to sewage sludge use regulations in Poland and Europe. It was also determined whether or not the treatment technology is critical in terms of the content of heavy metals in mobile forms. The importance of analyzing the form in which heavy metals are present became apparent when considering their use for agricultural purposes.

## 2. Materials and Methods

### 2.1. Characteristics of Collection Points and Potential Uses of Sludge

Sludge samples were collected from four different wastewater treatment plants located in the Swietokrzyskie province in Poland (Figure 1). The characteristics of the treatment plants are presented in Table 2. The plants differ in the type of sludge treatment, which are: oxygen stabilization, dewatering on belt press, Imhoff fermentation, and internal digester fermentation. Reference points for the content of heavy metals in soil were measuring stations prepared within the *Framework of Monitoring of Chemistry of Arable Soils in Poland* [23] located not far from the sampling points (Figure 1). Test Point 361 in Wola Kopcowa was selected as a point for potential use of sewage sludge. The soil was characterized by complex 2z (medium grassland), type: A (podzolic soil), and valuation class: IVb. The soil type was sandy loam. The “pH” in H_2_O suspension was 5.5, while in KCL, it was 4.5. The humus content of the soil was 3.24%, organic carbon 1.88%, total nitrogen 0.09%, while the C/N ratio was 20.89 [23].

### 2.2. Heavy Metal Speciation

Heavy metals can be classified into four mobility fractions based on their migration capacity [24]. These are:

FI fraction—associated with carbonates, the most mobile;FII fraction—associated with amorphous iron and manganese oxides;FIII fraction—associated with organic and sulfide matter;FIV fraction—associated with silicates—a completely chemically stable fraction.

The study used a four-step procedure developed by the European Community Reference Bureau, or BCR for short [25,26]:

Step I: CH_3_COOH extraction–(FI—exchangeable fraction);Step II: extraction NH_2_OH·HCl–(FII—reducible fraction);Step III: extraction H_2_O_2_/CH_3_COONH_4_–(FIII—oxidizable fraction);Step IV: mineralization of the residual fraction with a mixture of concentrated acids (HCl, HF, HNO_3_)–(FIV—residual fraction).

### 2.3. Heavy Metal Accumulation Risk Indicators

#### 2.3.1. Geoaccumulation Index of Heavy Metal in Soil (Igeo)

This specific method was proposed by Muller [27] to determine and classify the state of sludge/soil contamination at five levels, ranging from uncontaminated to highly contaminated. The geoaccumulation index (Igeo) measures the level of sediment or soil contamination by inorganic or organic trace substances of environmental concern and bioelements. It compares current concentrations to precivilization concentrations or, in the case of synthetic substances that do not occur in nature but have recently been produced, to reference values derived from an assumed uniform global distribution of these substances. Igeo is defined by the equation [27,28]:(1)$$\mathrm{Igeo}=log_{2}\frac{C_{n}}{1.5\cdot B_{n}}$$
where:

*C_n_*—the concentration of a specific heavy metal element in sewage sludge, mg∙kg^−1^ d.m.;*B_n_*—content of a given element from the group of heavy metals present in the soil, mg∙kg^−1^ d.m.

Table 3 presents the classification of the heavy metals geoaccumulation index and risk assessment code.

#### 2.3.2. Risk Assessment Code (RAC)

The risk assessment code (RAC) is a quantitative method for determining the mobility and bioavailability of heavy metals based on total metal concentration and chemical fraction. Because the acid-extractive fraction (F1), which consists of exchange fraction, has higher bioavailability, its mass fraction is used to evaluate metals in soils or sediments [28]. The RAC index introduced by Perin et al. [29], was classified into five risk categories (Table 4). It is calculated in accordance with [29,30]:(2)$$\mathrm{RAC}=\frac{F1}{\mathrm{HM}}\cdot 100\%$$
where:

F1—acid heavy metal concentration—soluble/free fraction; mg∙kg^−1^; HM—total heavy metal concentration, mg∙kg^−1^.

#### 2.3.3. Potential Environmental Risk Index (PERI)

The potential ecological risk index (PERI), developed by Hakanson (1980) [28], is based on the principles of sedimentology. Scientists use it extensively to evaluate the pollution and potential ecological risk associated with heavy metals in sewage sludge. This index takes into account not only the heavy metal content of sewage sludge, but also the ecological and environmental effects of heavy metals [30]. It is calculated by the following formulas [30,31,32]:(3)$$\;C_{f\;}^{i}=\frac{C_{D}^{i}}{C_{R}^{i}}$$
where:
$C_{f\;}^{i}$—pollution factor;$C_{D}^{i}$—concentration of the *i*-th element from the HM’s group present in sludge, mg∙kg^−1^ d.m.;$C_{R}^{i}$—concentration of the *i*-th element from the HM’s group in the soil, mg∙kg^−1^.
(4)$$E_{r}^{i}=T_{r}^{i}\;\cdot \;C_{f}^{i}$$
where:
$E_{r}^{i}$– index of the potential ecological risk of the *i*-th element from the HM group;$T_{r}^{i}$–toxicity factor of the i-th element from the HM group;The degree of heavy metal toxicity varies according to the toxicity factor. ($T_{r}^{i}$): Cd-30, Cu, Ni and Pb-5, and Zn-1 [30].

The sum of potential ecological risk from sludge in the ground is defined by the equation [31]:(5)$$\mathrm{PERI}=\sum_{i=1}^{n}E_{r}^{i}$$

The risk level is classified into 5 categories as shown in Table 5.

#### 2.3.4. Environmental Risk Determinant (ERD)

Considering the mobility of heavy metals, it can be seen that only Fraction IV does not migrate into the soil-water environment under any conditions. The mobile fractions (FI, FII) are considered to be the most mobile, while Fraction FIII can be mobile under certain conditions, i.e., when the organic matter in the soil is fully processed by microorganisms and when there is an ozone storm. Metals bound to iron and manganese oxides are released into the environment relatively slowly. Under certain conditions of pH and oxidation-reduction potential, metals bound to FII can exhibit significant bioavailability [33]. Environmental risk assessment is carried out based on the first three fractions, taking into account the level of individual predisposition of each fraction to release heavy metals into the soil environment. The ERD calculates the content of heavy metal elements based on their distribution in the four fractions. Each fraction is assigned a weight ranging from 0 to 1. The authors proposed using the ERD index because none of the indicators using the mobility issue take into account the weight of each fraction. Consider that the FI, FII, and FIII fractions are mobile, but the FI fraction is much more mobile than FII and FIII, which takes into account the formula for the ERD index. The adopted weight ranges were proposed based on the scale analysis of the other indicators. The content of metals found in Fraction I is taken into account in its entirety to determine the risk of ecological contamination, while Fractions FII and FIII, no longer completely mobile, were reduced by a procedure of potentiating the values to the second and third power, respectively. The applied scale ranges were proposed based on the evaluation of the scales of the other indicators. Its determinant is defined by the equation [33,34]:ERD=F_p1_ + F_p2_ + F_p3_(6)
where:

Fp_1_ = F_1_; F_1_—metal content in Fraction FI on a scale of 0–1; F_p2_ = F_2_^2^; F_2_—metal content in Fraction FII on a scale of 0–1; F_p3_ = F_3_^3^; F_3_—metal content in Fraction FIII on a scale of 0–1.

The risk level is classified into 4 categories as shown in Table 6.

## 3. Results

This section is divided into subsections that provide a concise description of the experimental results, their interpretation, and experimental conclusions. The results of chemical speciation of heavy metals in sewage sludge are shown in Table 7.

The Igeo index is largely dependent on the heavy metal content of the soil at the site of potential use. Because of this fact, a common site for sludge from all four WWTPs was chosen as a potential sludge use point. Figure 2 shows the Igeo value for all samples analyzed

For all analyzed cases of sewage sludge, the RAC showed no high risk of environmental contamination. This is due to the low content of heavy metals in the FI fraction. Statistically, sediments collected from WWTP4 have the highest percentage of metals in the FI fraction. The highest percentage was recorded for cadmium from WWTP4, which was 25%, but it did not reach the high-risk level. The outcomes can be considered satisfactory; however, it should be noted that the RAC index only considers the FI fraction, ignoring the heavy metals in the FII and FIII fractions. Figure 3 shows the RAC value for all sewage sludge samples.

Analyzing the PERI indicator for the studied sewage sludge samples, it can be concluded that copper, cadmium, and zinc are the main heavy metals that pose a threat to the environment. Other heavy metals showed low levels of potential environmental contamination. Figure 4 shows the PERI values.

The ERD index revealed similar risk levels to the RAC, but it was more accurate due to the inclusion of metals from fractions II and III. As with the RAC, sewage sludge collected from Wastewater Treatment Plant 4 proved to be the most hazardous in terms of potential metal migration. Figure 5 depicts the ERD values.

For the results of potential risk, for all indices and wastewater treatment plants, a noncompliance table was prepared (Table 8). The heavy metals listed in the table did not meet the criterion that would qualify the sludge for potential environmental use. As can be seen in Figure 2 and Figure 4, indicators comparing total sediment content to content at the point of use were far more critical than indicators considering mobility. Although the metal content of sediment is high, it may be in stable fractions that cannot migrate in soil or vegetation. As a result, it appears appropriate to consider the form in which the metal occurs when evaluating the possibility of natural sewage sludge use.

## 4. Discussion

This paper presents the analysis of sewage sludge taken from four treatment plants that use various water treatment technologies. The heavy metal content in all samples did not exceed the permissible metal content limit for agricultural use. However, the results of the analyzed indicators of contamination risk were not so encouraging. The most stringent proved to be the Igeo index, which compares the metal content of the sludge with the content in the geological substrate at the site of potential use. According to Igeo, all sediments posed a very high risk of ecological contamination, with zinc and cadmium proving particularly toxic. Another indicator considering metal mobility was RAC. According to the indicator value, the sediments showed a medium or low risk of contamination. This is related to their exchangeable fraction content. With respect to RAC, the most toxic sludge sample was the one taken from Treatment Plant 4: Heavy Metal. In all samples, cadmium and copper proved to be the most toxic and risky. The other metals showed medium or low levels of toxicity. The index proposed by the authors (ERD), which is based primarily on the issue of heavy metal mobility, identified WWTP 4 as having the highest risk of ecological contamination. The other three WWTPs, on the other hand, do not pose a high contamination potential risk. Copper is the heavy metal most likely to penetrate deep into the soil from the sludge samples of all analyzed wastewater treatment plants.

## 5. Conclusions

This paper examines the risk of environmental contamination caused by the agricultural use of sewage sludge from four wastewater treatment plants in Poland. Each of the analyzed facilities uses different wastewater treatment technology. Sewage sludge from all wastewater treatment plants met the applicable heavy metals limits imposed by legal acts, which is the primary criterion for using sewage sludge as a fertilizer. Conventional indices based solely on total heavy metal content, such as Igeo and PERI, were significantly more critical in assessing the feasibility of sludge use than the ERD and RAC, which also look at heavy metal chemical forms. As a result, most of the metals in the sludge were in a completely stable form, and despite their high concentration, there was no possibility of migration and entry into the crop. The study found that wastewater treatment technology has no significant impact on the total content of heavy metals in the sludge; however, sludge from a treatment plant that uses press dewatering has the highest content in the mobile sections. In all sewage sludge samples, copper was the most mobile heavy metal.

## Figures and Tables

**Figure 1 ijerph-19-11829-f001:**
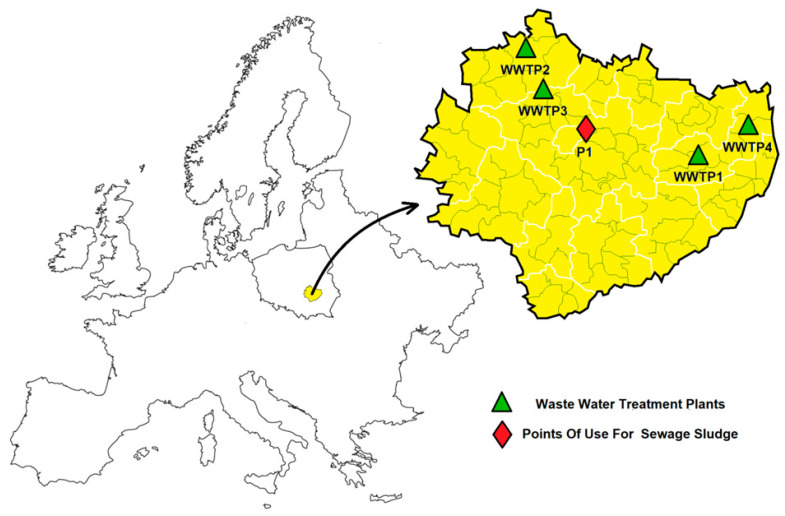
Location of WWTPs and potential sites of agricultural use of sewage sludge (own research).

**Figure 2 ijerph-19-11829-f002:**
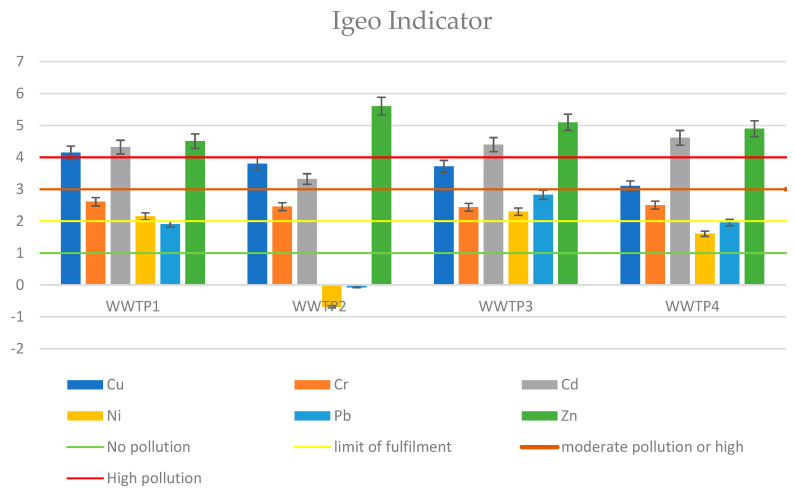
The geoaccumulation index (Igeo) of heavy metals in sewage sludge.

**Figure 3 ijerph-19-11829-f003:**
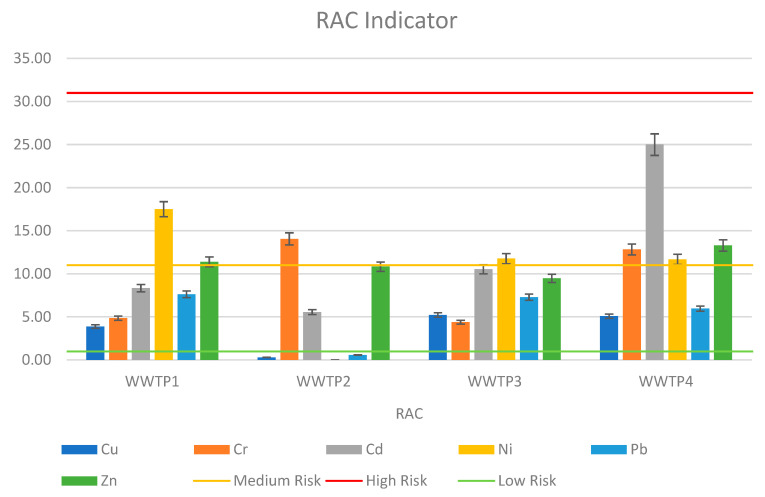
Risk assessment code (RAC) of HMs in sewage sludge.

**Figure 4 ijerph-19-11829-f004:**
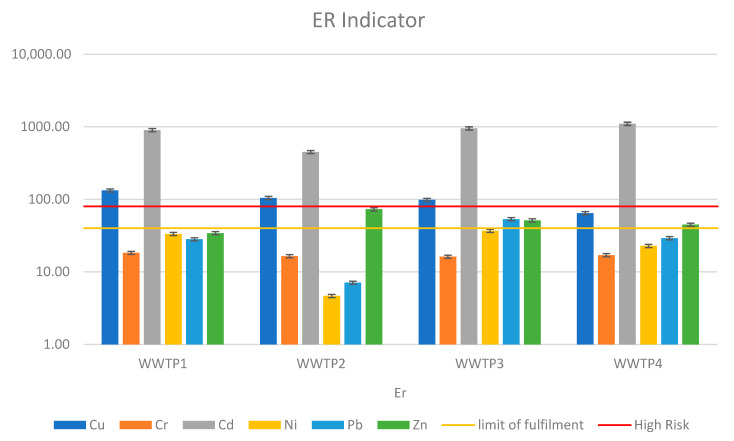
PERI indicator of heavy metals in sewage sludge.

**Figure 5 ijerph-19-11829-f005:**
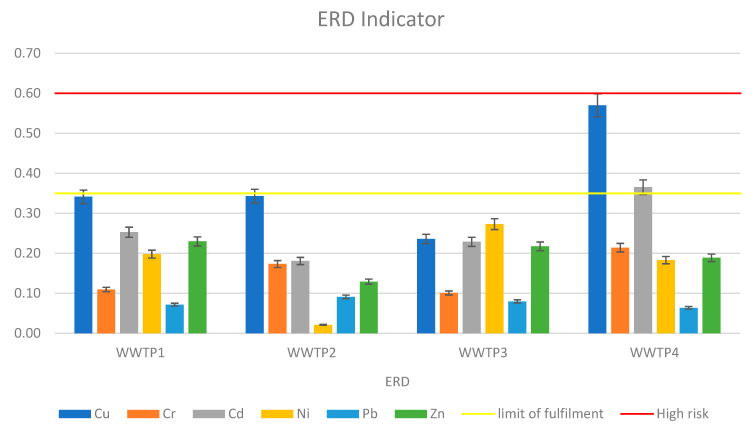
The environmental risk determinant (ERD) indicator of heavy metals in sewage sludge.

**Table 1 ijerph-19-11829-t001:** Heavy metal limit values in sewage sludge intended for natural use (mg/kg d.m.).

Metal	Limit Values for Heavy Metals in Sewage Sludge Intended for Natural Use					
	Poland Regulation [14]	EU Directive 86/278/EEC [15]	Chinese Regulation GB 18918-2002 [16]		USA Regulation 40 CFR Part 503, 503.13 [17]	South African Guideline (Pollutant Class a) [18]
			pH < 6.5	pH > 6.5		
Cd	20	20–40	5	20	39	40
Ni	300	300–400	100	200	420	420
Zn	2500	2500–4000	500	1000	2800	2800
Cu	1000	1000–1750	250	500	1500	1500
Cr	500	-	600	1000	-	1200
Pb	750	750–1200	300	1000	300	300

**Table 2 ijerph-19-11829-t002:** Characteristics of WWTPs (own research).

	Wastewater Treatment Plant			
	WWTP1	WWTP2	WWTP3	WWTP4
Location of WWTP	Opatow	Kornica	Mniow	Ozarow
Type of WWTP	Mech.-biol.	Mech.-biol.	Mech.-biol.	Mech.-biol.
Equivalent Number of Residents	15,240	21,594	9550	9660
SS treatment	Internal digester fermentation	Imhoff fermentation	Oxygen stab.	Dewatering on belt press
Distance of the WWTP from the point use of SS (km)	56	61	32	80

**Table 3 ijerph-19-11829-t003:** Classification of Igeo [28,29].

Igeo	Pollution Value
<0	No pollution
0–1	No pollution, moderate pollution
1–2	Moderate pollution
2–3	moderate pollution or high
3–4	High pollution

**Table 4 ijerph-19-11829-t004:** Classification of RAC [29,30,31].

RAC	Risk Value
<1	No risk
1–10	Low risk
11–30	Medium risk
31–50	High risk
>50	Very high risk

**Table 5 ijerph-19-11829-t005:** PERI indicator classification [30,31,32].

$E_{r}^{i}$	PERI	Risk Value
<40	<150	Low
40–80	150–300	Medium
80–320	300–600	High
>320	>600	Very high

**Table 6 ijerph-19-11829-t006:** ERD indicator classification [33,34].

ERD	Risk Value
0 < ERD ≤ 0.35	Low risk
0.35 < ERD ≤ 0.6	Medium risk
0.6 < ERD ≤ 0.8	High risk
0.8 < ERD	Very high risk

**Table 7 ijerph-19-11829-t007:** Chemical speciation of heavy metal in sewage sludge, for sludge from all four treatment plants, the results are the statistical average of four separate measurements for each sludge, excluding coarse errors, mg∙kg^−1^.

	Heavy Metal (mg/kg s.m.)					
Fraction	Cu	Cr	Cd	Ni	Pb	Zn
Sewage sludge—S1						
Fraction I	3.3 ± 0.2	2.0 ± 0.3	0.3 ± 0.1	3.5 ± 0.1	5.2 ± 0.1	79.4 ± 0.7
Fraction II	1.8 ± 0.1	1.1 ± 0.1	0.3 ± 0.1	1.4 ± 0.1	0.5 ± 0.2	122.8 ± 2.6
Fraction III	57.1 ± 1.1	16.1 ± 0.7	1.9 ± 0.1	5.9 ± 0.1	7.8 ± 0.1	323.8 ± 1.5
Fraction IV	22.8 ± 0.7	22.0 ± 0.7	1.1 ± 0.8	9.2 ± 0.3	54.7 ± 9.4	170.8 ± 1.3
ΣFI…IV	85.0 ± 2.3	41.2 ± 0.9	3.6 ± 0.9	20.0 ± 0.5	68.2 ± 11.3	696.8 ± 2.6
Sewage sludge—S2						
Fraction I	0.2 ± 0.1	5.2 ± 0.2	0.1 ± 0.1	0.0 ± 0.1	0.1 ± 0.1	161.3 ± 2.0
Fraction II	1.1 ± 0.1	0.8 ± 0.2	0.6 ± 0.1	0.4 ± 0.1	0.6 ± 0.2	71.7 ± 0.7
Fraction III	47.4 ± 0.8	12.1 ± 0.3	0.5 ± 0.1	0.3 ± 0.1	7.4 ± 0.8	356.0 ± 3.5
Fraction IV	18.3 ± 0.4	18.9 ± 0.4	0.6 ± 0.1	2.1 ± 0.3	9.1 ± 0.8	899.0 ± 9.2
ΣFI…IV	67.0 ± 0.9	37.0 ± 2.4	1.8 ± 0.5	2.8 ± 0.5	17.2 ± 3.1	1488 ± 8.0
Sewage sludge—S3						
Fraction I	3.3 ± 0.2	1.6 ± 0.1	0.4 ± 0.1	2.6 ± 0.2	9.4 ± 0.9	99.2 ± 2.2
Fraction II	1.6 ± 0.1	1.5 ± 0.1	0.3 ± 0.1	6.1 ± 0.5	11.1 ± 1.3	123.2 ± 2.9
Fraction III	36.1 ± 0.3	14.3 ± 0.3	1.7 ± 0.1	9.1 ± 0.7	9.9 ± 1.1	499.9 ± 8.2
Fraction IV	22.1 ± 0.3	19.1 ± 0.4	1.4 ± 0.1	4.3 ± 0.3	98.5 ± 9.9	324.5 ± 8.1
ΣFI…IV	63.1 ± 0.9	36.5 ± 0.5	3.8 ± 0.2	22.1 ± 0.7	128.9 ± 3.7	1047 ± 19.6
Sewage sludge—S4						
Fraction I	2.1 ± 0.2	4.9 ± 0.3	1.1 ± 0.1	1.6 ± 0.2	4.2 ± 0.5	121.2 ± 1.1
Fraction II	3.7 ± 0.1	0.1 ± 0.05	1.6 ± 0.1	3.8 ± 0.5	3.7 ± 0.4	111.2 ± 1.4
Fraction III	33.2 ± 0.2	16.1 ± 0.2	0.8 ± 0.1	0.2 ± 0.1	4.6 ± 0.5	329.0 ± 3.1
Fraction IV	2.3 ± 0.3	17.1 ± 0.5	0.9 ± 0.1	8.1 ± 0.6	58.0 ± 6.1	350.5 ± 2.1
ΣFI…IV	41.3 ± 1.2	38.2 ± 1.3	4.4 ± 0.6	13.7 ± 0.7	70.5 ± 11.7	911.9 ± 7.7

**Table 8 ijerph-19-11829-t008:** Schedule of failure to meet heavy metal toxicity criterion from analyzed sites for four pollutant indicators.

WWTP *	Igeo	RAC	PERI	ERD
WWTP1	Ni, Zn, Cu, Cr, Cd	Ni, Zn	Cu, Cd	-
WWTP2	Zn, Cu, Cr, Cd,	Cr	Cu, Cd, Zn	-
WWTP3	Ni, Zn, Pb, Cu, Cr, Cd,	Ni	Cu, Cd, Pb, Zn	-
WWTP4	Zn, Cu, Cd, Cr,	Cr, Cd, Ni, Zn	Cu, Cd, Zn	Cd, Cu

* WWTP1 (internal digester fermentation), WWTP2 (Imhoff fermentation), WWTP3 (oxygen stab.) WWTP4 (dewatering on belt press).

## Data Availability

Not applicable.

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
