# Peer review of "Risk Analysis of Heavy Metals Migration from Sewage Sludge of Wastewater Treatment Plants"

_ijerph, 2022, doi:10.3390/ijerph191811829_

Round 1

Reviewer 1 Report (Previous Reviewer 1)

The concept of the study is very innovative. However, the authors have not presented the data clearly to show how their objectives are full-filled. 

Please mention the context of the study clearly in the introduction. 

Present your data and interpret the  results how you have achieved your objectives. You did interpret the what the results of indices mean. 

Author Response

Dear Reviewer,

Thank you very much for reading and reviewing our work, and for your valuable suggestions. 
We have rewritten the introduction to make clear the context of the study, what our study aims to do, and how our research can be used. 
We included a discussion point and conclusions, reinterpreted the results, and referred to the stated objectives in the introduction. The indicators were interpreted and the results discussed. 

This manuscript is a resubmission of an earlier submission. The following is a list of the peer review reports and author responses from that submission.

Round 1

Reviewer 1 Report

The research idea or hypothesis of the manuscript is very good. However, writing and data presentation are poor. I would suggest to re-write the manuscript and submit again. 

For example, even the first line of the manuscript is grammatically incorrect: Environmental utilization receiving increased attention in sewage sludge management  processes. 

Please find some more specific comments:

1. Many  errors in sentence structure  which leads to misunderstanding - for example the first line of the abstract. 

2. I failed to find the link from abstract to conclusion: justification of the study, hypothesis/ goal of the study, clear methods for achieving the  objectives and results for specific objectives ( e.g. migration of metals).

3. I could not find the hypothesis/ goal of the study which we usually write at the end section of introduction. We need to show the existing knowledge and gaps on the topic.  

Therefore I suggest the authors re-shape or rewrite the manuscript and resubmit. 

Reviewer 2 Report

The paper “Risk analysis of heavy metals migration from sewage sludge of wastewater treatment plants” is of actual interest as sustainability is in the order of the day and environmental utilization is one of its main components.  The study is interesting and deserves publication. The authors have other published works on the subject.

However, the work is not ready for publication.

The abstract must be carefully checked as grammar faults prevent the understanding of the text. Also, the author must nor use ‘s in written language.

The introduction is too long and the author presents general subjects, jumps to very specific methodology details and then goes to general subjects again. So, this must be reorganized and some subjects can be allocated to the discussion.  

Materials and Methods Section must be checked for English language. Sentence of lines 140-150 must be re-written. Figure 1 can be simplified and the English must be checked as ‘s is also used and the graphical legend states “points of use for sewage sludge” and there is only one and the expression must be “of sewage sludge” as in the text legend. In line 160, the author states “The best results for sewage sludge samples were obtained using…”. What is the basis of this evaluation? Section 2.3.4 must be re-written as authors mix procedure guidelines with considerations about the various steps. They can do this but in a more straightforward way, clarifying what is the methodology and what are adaptations made by the authors of the work and reasons for this.

Results: The legend of Table 7 lacks important information. And that of Table 8 as well. The discussion of the results is relatively poor comparing, for example with the Introduction. The conclusions are not fully supported by the discussion of the results.

Reviewer 3 Report

Overall, an interesting study and a good experimental set up. The authors have done a good level of work in presenting the results, however, lack adequate discussions. I would recommend authors to revise their discussion section with added details on the observed experimental findings.

With the use of 4 different WWTPs and 4 different indices, it would be great if the authors can elaborate their discussion and make some conclusive statements based on discussions. Please discuss more on the indices selected for study and what would be the recommendations for other similar studies in the future based on your experimental observations.

Lastly, there were several linguistic and typo errors throughout the manuscript and authors are suggested for a proof-read of the article prior to re-submitting. For instance – several sentences in abstract, line 262-264, and others.

Reviewer 4 Report

See the attached review report.
